# Understanding the Molecular Interactions Between Influenza A Virus and *Streptococcus* Proteins in Co-Infection: A Scoping Review

**DOI:** 10.3390/pathogens14020114

**Published:** 2025-01-24

**Authors:** Askar K. Alshammari, Meshach Maina, Adam M. Blanchard, Janet M. Daly, Stephen P. Dunham

**Affiliations:** 1One Virology, Wolfson Centre for Global Virus Research, School of Veterinary Medicine and Science, Sutton Bonington Campus, University of Nottingham, College Road, Loughborough LE12 5RD, UK; alyaa24@nottingham.ac.uk (A.K.A.); meshach.maina@nottingham.ac.uk (M.M.); adam.blanchard@nottingham.ac.uk (A.M.B.); janet.daly@nottingham.ac.uk (J.M.D.); 2Department of Clinical Sciences, College of Veterinary Medicine, King Faisal University, Al-Hofuf 36388, Saudi Arabia

**Keywords:** influenza A virus, *Streptococcus* spp., co-infection, molecular interactions, neuraminidase, bacterial colonisation, viral–bacterial synergy, therapeutics

## Abstract

Influenza A virus infections are known to predispose infected individuals to bacterial infections of the respiratory tract that result in co-infection with severe disease outcomes. Co-infections involving influenza A viruses and streptococcus bacteria result in protein–protein interactions that can alter disease outcomes, promoting bacterial colonisation, immune evasion, and tissue damage. Focusing on the synergistic effects of proteins from different pathogens during co-infection, this scoping review evaluated evidence for protein–protein interactions between influenza A virus proteins and streptococcus bacterial proteins. Of the 2366 studies initially identified, only 32 satisfied all the inclusion criteria. Analysis of the 32 studies showed that viral and bacterial neuraminidases (including NanA, NanB and NanC) are key players in desialylating host cell receptors, promoting bacterial adherence and colonisation of the respiratory tract. Virus hemagglutinin modulates bacterial virulence factors, hence aiding bacterial internalisation. Pneumococcal surface proteins (PspA and PspK), bacterial M protein, and pneumolysin (PLY) enhance immune evasion during influenza co-infections thus altering disease severity. This review highlights the importance of understanding the interaction of viral and bacterial proteins during influenza virus infection, which could provide opportunities to mitigate the severity of secondary bacterial infections through synergistic mechanisms.

## 1. Introduction

Influenza A viruses (IAV) are a common cause of seasonal respiratory infections in humans and have the potential to cause pandemics. They are also important pathogens for a wide range of animals. They are divided into subtypes based on the two surface glycoproteins, hemagglutinin (HA), and neuraminidase (NA), with H1–H16 and N1–N9 all found in various combinations in aquatic birds. Human spillover led to the establishment of the two subtypes that continue to circulate with representative strains included in seasonal influenza vaccines, for example, an A/Victoria/4897/2022 (H1N1) and A/Thailand/8/2022 (H3N2). Similarly, certain IAV subtypes circulate endemically in swine and horses. IAV infection can also cause respiratory disease in both pigs and horses. Uncomplicated IAV infection is usually mild in pigs, but co-infection with other pathogens can cause severe disease. Pigs are highly susceptible to secondary bacterial infections during infection with IAV [1,2,3] and thus are a good model for studying the interaction between viral and bacterial pathogens. Pigs may also serve as a vessel for IAV evolution and the emergence of new strains, which can be zoonotic [1]. Understanding IAV-induced immune suppression and epithelial damage in pigs can provide important insights into the pathogenesis of these co-infections, which are pertinent to both human and veterinary health [1]. Influenza A virus infection is known to predispose infected individuals to bacterial infections. Particularly when novel IAV subtypes arise in a population, for example, during human pandemics, bacterial co-infection causes severe disease outcomes [4,5]. It is estimated that approximately 30% of cases during the 2009 H1N1 pandemic had bacterial co-infection with high mortality [6]. Similarly, secondary pneumonia from bacterial superinfections was associated with the majority of the estimated 50 million deaths that resulted from the 1918 influenza A virus (H1N1) pandemic [7,8]. The most common cause of bacterial pneumonia secondary to IAV infection in humans is *Streptococcus pneumoniae* (*S. pneumoniae*) [9,10]. Secondary bacterial infections in pigs and horses are similarly often associated with streptococcal infection (e.g., *S. suis* and *S. equi equi*, respectively).

During IAV replication, virus particles acquire a bilayered membrane by budding from respiratory epithelial cells, which causes damage, including initial clumping and subsequent erosion of cilia (Figure 1). This disruption to the mucociliary elevator paves the way for invasion by bacteria such as *S. pneumoniae* in human IAV infection to enhance the development of post-viral bacterial pneumonia [11]. However, the mechanisms of post-viral bacterial infection are multifaceted and are also mediated, in part, by molecular interactions between bacterial and viral proteins.

Virus surface glycoproteins play important roles in the virus life cycle, with HA mediating attachment and entry, and NA facilitating virus egress [12]. Influenza A viruses bind to epithelial cells in the respiratory tract via HA, targeting either α2,3- or α2,6-linked sialic acids [13,14]. Human influenza strains tend to preferentially bind to α2,6-linked sialic acids, found in the trachea, while avian-like strains bind more efficiently to α2,3-linked sialic acids, predominantly expressed on alveolar cells in human lung [13,15,16]. Once inside the host cell, viral RNA activates immune pathways, such as the retinoic acid-inducible gene I (RIG-I) pathway and the inflammasome, triggering pro-inflammatory cytokines [17,18,19]. While these immune responses aim to clear the virus, the viral proteins can also cause apoptosis and necrosis of epithelial cells, leading to tissue damage [20,21]. This epithelial damage, in turn, facilitates bacterial colonisation by providing exposed surfaces and damaged barriers for bacterial invasion [19,22]. Viral NA cleaves sialic acid residues on host cell surfaces, facilitating the release of newly formed virions [18,23,24].

Viral NA plays a crucial role in IAV replication and spread [25,26,27]. NA, a surface glycoprotein of the influenza virus, cleaves sialic acid residues from glycoproteins and glycolipids on the surface of infected host cells [23,28]. This enzymatic activity allows newly formed viral particles to exit infected cells and disseminate throughout the respiratory system. Without this function, virions would remain attached to the host cell surface, limiting viral spread to the initially infected cells [29].

*Streptococcus* spp. have various surface proteins, such as M protein and pneumococcal surface proteins (PspA and PspK), that bind to epithelial tissues, modulate the immune system and evade immune responses [30,31]. *S. pneumoniae* isolates express neuraminidase proteins (NanA, NanB and NanC), all of which, like the IAV NA, cleave the terminal sialic acid of glycan structures on host cell surfaces. This desialylation process exposes underlying glycoprotein and glycolipid receptors, creating adhesion sites that enhance bacterial attachment and promote biofilm formation—a critical step in establishing persistent infections. In complex co-infections, the released sialic acids are also broken down for metabolic use as a carbon source by bacteria, while modifications to bacterial surface molecules (which for gram-negative bacteria include lipopolysaccharides (LPS)), help bacteria evade the host’s immune system, form protective biofilms, and resist environmental stresses [32,33]. Other Streptococcal species, such as *S. pyogenes* and *S. suis*, though lacking NA, possess proteins such as streptolysins (SLO, SLS), which may play similar roles by disrupting host cell membranes and promoting bacterial colonisation [34,35,36,37].

During viral co-infections, the host’s immune defences are “weakened”, as the immune system is primarily focused on fighting the viral infection. This immune suppression or “distraction” reduces the host’s ability to effectively detect and respond to secondary bacterial invaders. Additionally, viral infections often lead to the release of pro-inflammatory cytokines, which can damage epithelial barriers and create a more accessible environment for bacterial adherence and invasion. In this altered state, bacteria find it easier to colonise and establish stable colonies, taking advantage of both exposed adhesion sites and the host’s reduced capacity for immune surveillance and response [19,38,39].

The aim of this review is to summarise the current understanding of the interactions between IAV and Streptococcus species during co-infection, focusing on the role of viral and bacterial proteins. We outline the role of these interactions in disease pathogenesis and how they contribute to severe disease outcomes and potential therapeutics.

## 2. Methods

### 2.1. Search Strategy

Three electronic databases: PubMed, CAB Abstracts, and Google Scholar were searched. For Google Scholar, Harzing’s Publish or Perish tool version 8.16.4764.9054 (https://harzing.com/resources/publish-or-perish, accessed on 1 July 2024) was used to extract relevant studies. The literature search employed Medical Subject Headings (MeSH) terms and Boolean operators (‘AND’ and ‘OR’), using the following keywords: (Infection OR co-infection OR superinfection OR superinfecting OR secondary infection OR enhances OR synergism) AND (influenza A OR equine influenza virus OR influenza) AND (streptococcus equi OR Streptococcus). No language restrictions were applied to the search terms across all fields. In PubMed, a filter was applied to limit the search to studies where specific terms appeared in the title or abstract, using the following search string: (Infection OR co-infection OR superinfection OR superinfecting OR secondary infection OR enhances OR synergism) AND (influenza A[Title/Abstract] OR equine influenza virus[Title/Abstract] OR influenza[Title/Abstract]) AND (streptococcus equi[Title/Abstract] OR Streptococcus[Title/Abstract]). Initially, the focus was on co-infection of equine influenza virus and *Streptococcus equi*, but due to the low number of studies available, the search was expanded to include broader terms related to influenza A and Streptococcus species. The results were entered into a Microsoft Excel (2019) file for screening based on inclusion and exclusion criteria. The reference lists of the selected articles were reviewed to identify additional relevant studies (Figure 2).

### 2.2. Eligibility Criteria and Data Extraction

Studies conducted worldwide between 1 January 1980 and 1 July 2024, were assessed for inclusion based on predetermined eligibility criteria. Eligible studies primarily focused on co-infections or superinfections with analysis of protein interactions between IAV and *Streptococcus* species, with experiments conducted in animals or cell models. Research investigating potential mechanisms underlying direct and indirect interactions between the IAV and *Streptococcus* species was also included.

The exclusion criteria included studies not published in English, review articles, case reports, case series, and studies involving human subjects (unless cell culture or animal models were used). This was conducted to ensure that the focus of the review remained on mechanistic insights, specifically on protein interactions and related therapeutic or immune mechanisms. Studies that exclusively reported clinical outcomes without examining these molecular processes were excluded, as they did not align with the objective of this review, which was to explore the underlying biological mechanisms rather than clinical presentations. Additionally, vaccine studies, conference posters, and book chapters were considered ineligible as they did not typically provide the primary research data needed for in-depth analysis of these mechanisms. To ensure the inclusion of high-quality and relevant studies, strict adherence to these criteria was essential to preserve the reliability of the findings. Any deviation from these criteria could have introduced bias, undermined the study’s validity, or resulted in the inclusion of studies with limited methodological rigour. For example, studies with poorly defined endpoints or inappropriate populations could have weakened the strength of the conclusions drawn.

Studies with titles and abstracts that did not meet the eligibility criteria were excluded, as were studies with missing or duplicate data from already included studies. When eligibility could not be determined from the title and abstract alone, full-text articles were retrieved for further review. Retained full-text articles were screened for eligibility based on the inclusion and exclusion criteria, and final studies were selected for data extraction. Two authors (AA and MM) performed independent identification, screening, and eligibility assessments and then compared the final selection of studies. Any discrepancies were resolved through discussion or, if necessary, by consulting other reviewers, either AMB, JMD, or SPD.

Finally, the extracted data from each study were organised and categorised into the following fields: Study ID, authors, year of publication, viruses studied, bacterial strains studied, experimental model (e.g., animal or cell culture), proteins studied (e.g., viral or bacterial proteins), type of interaction examined (e.g., direct or indirect), and the key findings. This categorisation enabled systematic comparison across the studies and facilitated the identification of common themes, gaps in the literature, and key mechanisms in the interaction between IAV and *Streptococcus* species.

## 3. Results

### 3.1. Study Features

Table 1 provides an overview of the key characteristics of the 32 studies analysed. Across the 32 studies analysed, four subtypes of IAV were used: H1N1, H3N2, H2N2, and H9N2, with one study only mentioning IAV without specifying a subtype. The most frequently identified subtype, H1N1, was found in 20 studies, of which 14 specifically used the cell-culture and mouse-adapted H1N1 laboratory strain A/Puerto Rico/8/34 (PR8). Eleven studies used an H3N2 subtype with the A/Hong Kong/1/68 (H3N2) isolate used in six of those studies. The H9N2 and H2N2 subtypes were used less frequently; they were mentioned in one study and three studies, respectively.

Of the 32 studies, in terms of the bacterial isolate used, *S. pneumoniae* appeared in 21, 10 studies used *Streptococcus pyogenes* (*S. pyogenes*), *Streptococcus suis* (*S. suis*) was used in two studies, while Group B *Streptococcus* appeared in one study. The presence of other bacteria, such as *Staphylococcus aureus* (four studies), *Neisseria meningitidis* (one study), and *Haemophilus influenzae* (two studies), was also found along with one of the *Streptococcus* species. In studies of bacterial-viral interactions, *S. pneumoniae* and *S. pyogenes* were most often described (Table 1).

In co-infection, several viral proteins were examined for their interaction with bacterial proteins. A total of 14 studies (43.75%) investigated NA, while 12 studies (37.5%) investigated HA (Table 1). Nucleoprotein (NP), polymerase basic 1 (PB1), PB2, and non-structural protein-1 (NS1) were each examined in one or two studies.

In terms of bacterial proteins, bacterial NA (NanA, NanB, and NanC) was the most extensively studied, appearing in 11 studies (34.4%). Pneumococcal surface proteins (PspA and PspK) and M protein were examined in five studies (15.6%), while pneumolysin was analysed in four studies (12.5%). Additional proteins, such as streptolysin O (SLO) and streptolysin S (SLS), appeared in three studies (9.4%). Other proteins, streptococcal pyrogenic exotoxin A (SPEA), hyaluronidase (Hyl), and fibronectin-binding proteins featured in one or two studies (Table 1). In addition, capsular sialic acid and hyaluronic acid were featured in one or two studies.

Regarding experimental models, a total of 21 studies (65.6%) used an in vivo mouse model. Nine studies (28.1%) used the A549 human lung cell line. In four studies (12.5%), MDCK cells were used, whereas other models, such as HEp-2 cells and porcine tracheal cells, were used in one study. Three studies (9.4%) did not use any specific model (N/A). Viral attachment, bacterial colonization, and host-pathogen interactions have been studied using these models.

Finally, 27 studies (84.4%), reported indirect interaction(s) between viral and bacterial proteins. As a result of these interactions, viral proteins such as NA and HA modified the host environment in such a way that they allowed bacteria to colonise, adhere, and evade the immune system. However, five studies (15.6%) reported direct interactions between viral proteins and bacterial proteins, such as the HA protein or streptococcal pyrogenic exotoxin A (SPEA). These interactions intensified the co-infection by promoting bacterial entry into host cells and worsening the infection.

The studies included in this review examined a variety of infections resulting from viral and bacterial interactions. While many studies focused on respiratory infections, such as pneumonia and influenza, some also investigated non-respiratory infections, including necrotizing fasciitis and other systemic infections including bacteraemia. The route of inoculation varied, with intranasal administration being common in animal models to mimic natural infection pathways. Co-infections, defined as concurrent infections caused by two or more pathogens, can involve either pathogens from the same category (e.g., multiple viruses) or from different categories (e.g., viruses and bacteria), and are a significant cause of increased morbidity and mortality compared with uncomplicated infections. In this review, the term co-infection specifically refers to cases where viral and bacterial pathogens interact to cause disease.

### 3.2. Bacterial Proteins

Most of the studies included in this review (n = 23) highlight several bacterial proteins that play critical roles in facilitating bacterial colonisation and pathogenesis during viral–bacterial co-infections. Notably, multiple studies (n = 11) focused on bacterial NA proteins (NanA, NanB, NanC) due to their essential role in cleaving sialic acid residues from host epithelial cells, enabling bacterial adhesion and colonisation (see Section 3.4).

Additionally, pneumococcal surface proteins (PspA, PspK) have been identified as important mediators of bacterial attachment to virus-infected host cells. These proteins facilitate *S. pneumoniae* adherence to epithelial cells by binding to host proteins such as GAPDH, establishing stable interactions that promote bacterial colonisation during viral co-infections. PspA and PspK not only aid bacterial binding but also play a key role in the dissemination of bacteria to other parts of the body, thereby worsening infection severity [42,59]. In particular, PspA was found to enhance colonisation of the nasopharynx and lungs through influenza co-infections, promoting bacterial spread and exacerbating pneumonia.

The M protein, primarily associated with *S. pyogenes*, is another critical bacterial adhesion protein extensively studied for its role in bacterial virulence. The M protein increases bacterial adhesion to host epithelial surfaces by binding to host proteins such as albumin, fibronectin, and fibrinogen. During co-infections with influenza viruses, this interaction facilitates *S. pyogenes* infiltration into host tissues. Research by Hafez, Abdel-Wahab et al. [34] and Herrera, Suso et al. [35] demonstrated that the M protein plays a dual role in bacterial adhesion and internalisation into host cells, which can intensify infection severity, compromise lung barrier function, and increase mortality rates in individuals with viral–bacterial co-infections.

Pneumolysin (PLY), a pore-forming toxin produced by *S. pneumoniae*, was also found to play a key role in exacerbating co-infections [33,53]. PLY disrupts host epithelial barriers, facilitating bacterial invasion and modulating immune responses to favour bacterial survival. According to Wolf, Strauman et al. [49], PLY impairs the function of immune cells, contributing to immune evasion and more severe lung damage during co-infections.

Other bacterial proteins identified for their roles in enhancing bacterial adhesion and pathogenesis during co-infections include streptolysin O (SLO) and streptolysin S (SLS), both of which are toxins produced by *S. pyogenes.* These proteins facilitate bacterial dissemination and tissue invasion by disrupting host cell membranes and promoting bacterial survival.

In summary, aside from NA, several bacterial proteins including pneumococcal surface proteins, M protein, and pneumolysin play crucial roles in enhancing bacterial adhesion, colonisation, and survival in the host during co-infections with influenza viruses.

### 3.3. Role of Viral Hemagglutinin and Non-Structural Proteins

The role of viral HA in enhancing bacterial co-infections was determined in 12 studies. Of the 12 studies, five studies [36,42,43,45,52] demonstrated direct interactions in which viral HA binds directly to bacterial components, such as sialic acid on the bacterial surface (Figure 3). The remaining seven studies [26,27,28,33,34,36,55] focused on indirect interactions, when the host environment is altered by the viral HA to facilitate bacterial colonisation. This includes mechanisms such as promoting the removal of sialic acid, which exposes underlying receptors on host cells, such as galectin receptors, thereby facilitating bacterial adherence (Figure 3) [27].

The viral HA facilitates the internalisation of *S. pyogenes* into epithelial cells, as demonstrated in animal models where prior IAV infection enhanced bacterial invasion and promoted lethal outcomes [36]. Similarly, HA enhances the adhesion of influenza virions to Group B Streptococcus, with specific bacterial serotypes binding to the viral receptors, thereby increasing adherence and interaction with host cells [43].

A study by Wu, Meng et al. [52] revealed a direct interaction between viral HA and *Streptococcus suis*, with the former binding to the sialic acid on the bacterial capsule. This interaction promotes bacterial adherence and invasion of pig tracheal epithelial cells. Interestingly, while the viral HA facilitates bacterial colonisation, the co-infection suppresses viral replication, as evidenced by delayed virus growth and reduced virus titres in infected cells.

How HA indirectly influences bacterial adherence through interactions with host proteins was also explored in studies. HA enhances galectin protein binding to pneumococci, which facilitates *S. pneumoniae* adherence to desialylated airway epithelium [28]. According to Walther, Xu et al. [27] during co-infections, bacterial NA proteins accelerate bacterial colonisation and increase viral replication through removing sialic acids from host cell surfaces along with viral HA binding to these receptors [27]. Furthermore, viral HA modulates the growth and virulence of *Streptococcus pneumoniae*, influencing capsule production and virulence factors, complicating the course of co-infections [33].

Several of the studies explored how HA synergises with bacterial NA or modulates host cell factors to create a more favourable environment for bacterial colonisation [26,34,43]. Ortigoza, Blaser et al. [26] demonstrated how HA, together with bacterial NA (NanA and NanB), modulates both viral transmission and bacterial colonisation. HA increases the interaction between *S. pyogenes* and MDCK cells, enhancing bacterial internalisation [34]. Another study showed that the swine influenza HA binds directly to sialic acid on *Streptococcus suis*, promoting bacterial adhesion and invasion into tracheal epithelial cells [43].

Klonoski, Watson et al. [55] demonstrated that the H1N1 subtype induces a stronger and prolonged proinflammatory cytokine response, leading to prolonged bacterial presence in the lungs and systemic spread, especially in cases of *S. pyogenes* infections, which correlates with higher mortality during bacterial superinfection. While the HA expressed by the virus is implicated in modulating lung inflammatory responses, the study also highlights the role of viral NS1 in shaping the cytokine environment and influencing superinfection severity. In contrast, there was more rapid viral clearance in H3N2 infection and less severe bacterial superinfections due to reduced inflammation and faster pathogen resolution [51,55]. These findings suggest that both HA and NS1 contribute to the observed stronger proinflammatory cytokine response in H1N1 infections and its association with worsened outcomes.

The PB1-F2 protein, an accessory protein encoded by IAV, plays a significant role in exacerbating secondary bacterial infections. It promotes immune cell apoptosis and modulates proinflammatory responses, leading to increased infiltration of immune cells such as neutrophils and monocytes. The protein has been shown to impair macrophage function, reducing bacterial clearance and enhancing bacterial survival and replication in the lungs. Specifically, PB1-F2 expression, particularly the full-length forms observed in virulent strains, has been linked to severe bacterial superinfections, such as those caused by S. pneumoniae and S. pyogenes. The presence of virulent PB1-F2 forms correlates with higher lung bacterial titres and mortality rates during co-infections, highlighting its pivotal role in synergistic viral–bacterial pathogenesis [46,54].

### 3.4. Role of Bacterial and Viral Neuraminidases

Both influenza viruses and bacteria such as *S. pneumoniae* rely on NA as a key enzyme in viral–bacterial co-infections. From the 32 studies analysed, NA was one of the most frequently studied proteins, with 14 studies (43.7%) focusing on viral NAs and 11 studies (34.4%) investigating bacterial NAs (NanA, NanB, NanC). Of these, five studies (35.7%) examined bacterial neuraminidases NanA, NanB, and NanC together, highlighting how these enzymes collectively contribute to bacterial colonisation and adherence by cleaving sialic acid on host cells. While viral and bacterial NAs share the ability to cleave sialic acid residues, their specific roles in co-infections differ, highlighting their distinct yet synergistic contributions to disease progression.

In a total of 14 studies, the role of viral NA in viral and bacterial co-infections was comprehensively examined. Seven of these studies specifically investigated how viral NA enhances bacterial adherence by exposing host cell receptors to which bacteria can bind. Viral NA activity significantly increased *S. pneumoniae* adherence to host cells, with higher NA activity correlating with more severe pneumococcal colonisation and increased mortality in co-infected mice. Specifically, the viral NA of the A/Singapore/1/57 (Sing57) strain showed a 3.8-fold increase in pneumococcal adherence, while the viral NA of the A/Sydney/5/97 (Syd97) strain exhibited a 2.9-fold increase. Moreover, viral NA activity also resulted in higher mortality rates. Mice infected with the Sing57 strain had a 42% mortality rate, while those infected with the Syd97 strain had a lower mortality rate of 21% [58]. Similarly, a lethal synergism between IAV and *S. pneumoniae* was observed, with 100% mortality occurring when pneumococcal infection followed influenza by seven days, leading to severe pneumonia. This time-dependent effect showed the highest mortality at this interval. The study investigated the role of the platelet-activating factor receptor (PAFr), hypothesising that its upregulation during influenza infection might enhance pneumococcal adherence and invasion. However, inhibiting PAFr had no significant impact on survival, indicating that the synergism may involve PAFr-independent mechanisms [51]. Both studies focused on *Streptococcus pneumoniae*, a bacterial species that also produces NA [51,58].

Building on these results, viral NA also exposes host cell bacterial receptors, increasing the adherence of several pathogens, such as *S. pneumoniae*, *Staphylococcus aureus*, and *Neisseria meningitidis*, and consequently causing severe secondary infections [60] (Table 1). Furthermore, it was demonstrated by Li, Ren et al. [24] that viral NA activates host cell signalling pathways, particularly the TGF-β pathway, which results in the upregulation of fibronectin and other host receptors and facilitates bacterial binding. Enhanced bacterial adhesion during co-infections is largely dependent on the activation of host cell signalling pathways by viral NA, particularly the TGF-β pathway, which results in the upregulation of fibronectin and other host receptors. Additionally, viral NA modulates the growth and capsule production of *S. pneumoniae*, which aggravates bacterial infections during influenza and influences bacterial virulence [33,51]. Moreover, viral NA was shown to facilitate pneumococcal adherence in both mouse models and in vitro A549 cell cultures by desialylating epithelial cells, exposing galactosyl moieties that are recognised by host proteins galectin-1 (Gal1) and galectin-3 (Gal3). These galectins act as molecular bridges, enhancing the binding of *S. pneumoniae* to the epithelial cells, thereby promoting bacterial adhesion. These findings underscore the crucial role of viral NA in modifying host cell surfaces, increasing susceptibility to secondary bacterial infections during influenza, which is a key mechanism in post-influenza bacterial pneumonia [28].

In addition to their role in bacterial adhesion, the synergistic effects of viral and bacterial NA were investigated in three of the 14 studies [23,26,47]. Both viral NA and bacterial NA (NanA, NanB, and NanC) co-operatively desialylate host cells [23], promoting bacterial colonisation more effectively than either enzyme alone. Viral NA and bacterial NA (NanA) worked together to increase nasal colonisation and exacerbate the severity of middle ear infections in the mouse model [47]. Specifically, Viral NA facilitated bacterial adhesion to epithelial surfaces, and bacterial NA increased infection intensity, demonstrating a collaborative role between them. Using an infant mouse model, Ortigoza, Blaser et al. [26] investigated how bacterial and viral NA alter host cell sialic acid residues and influence the spread of IAV. Their study revealed that pneumococcal sialidases, such as NanA and NanB, play a critical role in modulating IAV transmission by removing sialic acid residues from host cells in the upper respiratory tract (URT). This depletion of sialic acids limits IAV’s ability to infect and spread to other cells, thereby reducing viral acquisition and transmission. The findings suggest that bacterial colonisation of the URT by *S. pneumoniae* can antagonize IAV transmission, highlighting the complex interactions between bacterial sialidases and viral transmission dynamics. Additionally, the study demonstrated that viral shedding is a key determinant of transmission, with both viral and host factors (such as age, immune response, and URT microbiota) influencing transmission efficiency.

Finally, 13 out of the 14 studies (Table 1) that examined the function of viral NA in co-infections with bacteria focused on indirect interactions, in which the NA of the virus promotes bacterial colonisation by cleaving sialic acids from host cell surfaces, thereby exposing receptors to which bacteria can attach. These studies consistently demonstrated that viral NA plays a crucial role in promoting bacterial adherence by modifying the host environment rather than directly interacting with bacterial cells. Nonetheless, a study conducted in 1980 by Sanford, Smith et al. [42] showed a direct link by demonstrating that the cleavage of host sialic acids by viral NA facilitates the attachment of virions to *Streptococci*, thereby directly increasing bacterial adhesion. This process was blocked by receptor-destroying enzymes, confirming the direct role of viral NA in mediating bacterial attachment.

Studies by Wren, Blevins et al. [47] and Klenow, Elfageih et al. [20] demonstrated that viral and bacterial NAs synergise during co-infections to promote increased bacterial adhesion and colonisation of respiratory epithelial cells. Wren, Blevins et al. [47] used an in vivo mouse model, where they infected mice with IAV followed by S. pneumoniae. They found that NanA-deficient pneumococci significantly reduced colonisation in the nasopharynx and middle ear compared to wild-type strains, indicating that NanA enhances bacterial adherence and infection in these respiratory tissues during viral co-infection. Klenow, Elfageih et al. [20] conducted a comparative enzymatic analysis using recombinant neuraminidases from influenza virus (N1 and N2) and S. pneumoniae (NanA, NanB, and NanC). They tested these enzymes on synthetic and multivalent sialic acid-containing substrates to mimic host glycoproteins and glycolipids, observing that viral NA initially cleaved sialic acid residues and bacterial NAs furthered this desialylation process. This desialylation process exposes underlying glycan structures on host respiratory epithelial cells, providing new adhesion sites for bacterial attachment. Klenow, Elfageih et al. [20] demonstrated this mechanism by quantifying enzymatic activity on synthetic substrates designed to mimic host cell surfaces, while Wren, Blevins et al. [47] confirmed the relevance of these findings in vivo by comparing bacterial load and tissue colonisation in NanA-expressing versus NanA-deficient bacterial strains. This co-operation exacerbates secondary bacterial infections, particularly with S. pneumoniae. In some instances, however, bacterial NAs may antagonize viral transmission by reducing the availability of host cell receptors [26].

### 3.5. Treatments

Four of the 14 studies (Table 1) examined NA inhibitors for their potential to limit bacterial colonisation and reduce the spread of influenza during co-infection. Compounds such as diazenylaryl sulfonic acids and pyrrolo[2,3-e]indazole, identified by Hoffmann, Richter et al. (2017) and Egorova, Richter et al. (2023), have demonstrated the ability to inhibit both viral and bacterial neuraminidases (NAs), reducing bacterial colonisation and viral replication. These inhibitors function as sialic acid analogues, blocking neuraminidase activity and disrupting the synergistic interaction between IAV and *S. pneumoniae*. For example, diazenylaryl sulfonic acid has shown significant potential by simultaneously reducing viral replication and bacterial adhesion during co-infections, while pyrrolo[2,3-e]indazole effectively targets viral NA (H1N1 and H3N2 strains) and pneumococcal NA (NanA). Additionally, well-established antivirals such as oseltamivir have proven effective in reducing the severity of co-infections by inhibiting viral NA, as highlighted in both clinical and in vitro studies [27,44,51].

However, the widespread clinical use of oseltamivir has led to resistance in certain influenza strains, reducing its efficacy and underscoring the need for alternative therapeutic agents [27]. Dual-action inhibitors, which target both viral and bacterial NAs, represent a promising strategy to overcome these limitations [50,57]. By preventing sialic acid degradation and reducing bacterial colonisation, these inhibitors disrupt the cooperative mechanisms that drive viral–bacterial synergy [57]. While current NA inhibitors, such as oseltamivir, zanamivir, peramivir, and laninamivir, are approved for clinical use and provide effective antiviral action [61], alternative dual-action compounds such as diazenylaryl sulfonic acids and pyrrolo[2,3-e]indazole have shown promising results in vitro [50]. Nevertheless, these compounds have yet to progress to human clinical trials, highlighting the need for further research and development to evaluate their potential as viable clinical treatments for severe viral–bacterial co-infections [50,57].

## 4. Discussion

This review highlights the critical role of both viral and bacterial proteins in driving the progression and severity of viral–bacterial co-infections. The interactions between influenza A viruses and *Streptococcus* species during co-infections are complex, involving a range of synergistic mechanisms that promote bacterial colonisation, enhance viral replication, and worsen disease outcomes. Based on the 32 studies analysed, these protein interactions are pivotal in shaping the pathogenesis of co-infections and offer insights into potential therapeutic strategies to mitigate their impact. In addition to viral NA and HA, this review highlights not only the extensively studied bacterial neuraminidase proteins (NanA, NanB, and NanC) but also other key *S. pneumoniae*-related proteins such as PLY, PspA, and PspK. These proteins are critical in facilitating bacterial adhesion, colonisation, and immune evasion during viral–bacterial co-infections. Additionally, the role of proteins such as M protein (*S. pyogenes*), streptolysin O, and streptolysin S demonstrate that similar virulence mechanisms exist across different *Streptococcus* species, highlighting potential therapeutic targets for diverse bacterial pathogens.

Neuraminidase is one of the most extensively studied proteins in the context of co-infections, reflecting its central role in these dynamics. Of the studies reviewed, 14 (43.75%) investigated viral NA, while 11 (34.38%) focused on bacterial NA (NanA, NanB, and NanC). Desialylation by NA not only aids in viral dissemination but also creates a more favourable environment for bacterial colonisation, potentially by exposing host receptors. However, the exact nature of these receptors remains largely unknown, highlighting an important area for future research.

The synergistic interaction between viral and bacterial NA was a key finding in several studies. These studies demonstrated that the combined activity of viral and bacterial NAs accelerates desialylation, which facilitates bacterial colonisation and significantly worsens secondary bacterial infections. Although this review focuses on non-human cases and excludes human studies, similar mechanisms have been documented in human infections. Specifically, studies on human co-infections have shown that viral and bacterial NAs cooperate to enhance bacterial colonisation and exacerbate disease severity, including middle ear infections [62]. However, our focus on animal studies aimed to provide a targeted understanding of these interactions in non-human models, as they offer unique insights into the dynamics of viral and bacterial protein interactions under experimental conditions. In certain instances, bacterial NA may play a more complex role, such as antagonizing viral transmission by limiting the availability of host cell receptors [26]. Despite this, the prevailing evidence suggests that viral and bacterial NA typically cooperate, amplifying the severity of co-infections by promoting both viral spread and bacterial adherence.

The viral HA was identified as the second most extensively studied viral protein in this review, with 12 investigations highlighting its role in co-infections. Pre-existing bacteria, such as *S. pneumoniae*, can further exacerbate influenza severity by secreting proteases that activate HA or by stimulating host proteases such as plasminogen, enhancing viral replication and infectivity.

The review also identified several bacterial proteins that are critical for bacterial adhesion and colonisation during co-infections. In several studies, PspA and PspK have been identified as mediators of pneumococcal attachment to host cells. They bind to host factors, such as GAPDH, which facilitate bacterial colonisation, especially in the respiratory epithelium during co-infections with influenza. In addition, *S. pyogenes* M protein enhances bacterial adhesion and internalisation by binding to host proteins, including fibronectin and fibrinogen. The presence of these bacterial adhesion proteins contributes to the severity of co-infections because they enable bacteria to establish themselves more effectively during viral infections.

The studies reviewed employed a variety of experimental models, each with its own strengths and limitations. In vivo models, particularly mouse models, are frequently used to study co-infection and immune response dynamics. While mice provide valuable insights into immune system interactions and can be genetically manipulated, they are not natural hosts for influenza A virus, which can limit the translatability of findings to humans. Additionally, mouse immune responses and disease pathology may not fully replicate those seen in humans. However, the ability to evaluate complex immune interactions, such as the role of macrophages and cytokine responses during viral and bacterial co-infections, remains a key advantage of using in vivo models. In contrast, in vitro models, such as cell cultures, allow for controlled studies of viral–bacterial interactions at the molecular level but lack the complexity of an intact immune system. These models are useful for studying specific protein interactions and the initial stages of infection but do not capture the full spectrum of immune-mediated effects seen in in vivo systems. Thus, both models contribute to understanding co-infections, but it is important to interpret their findings with an awareness of these limitations.

Therapeutically, the potential of targeting NA in co-infections has been highlighted in several studies. Oseltamivir, a widely used antiviral for the clinical treatment of influenza infections, has been shown to reduce the severity of co-infections by inhibiting viral neuraminidase. However, while oseltamivir has been successfully used in human clinical cases, its widespread use has led to the emergence of resistance, particularly in certain influenza strains, reducing its efficacy [27]. This resistance underscores the urgent need for alternative therapeutic agents with enhanced efficacy and reduced susceptibility to resistance. Dual-action inhibitors such as diazenylaryl sulfonic acids and pyrrolo[2,3-e]indazole compounds, which target both viral and bacterial neuraminidases, represent a promising avenue for overcoming these limitations, particularly in the context of severe viral–bacterial co-infections following influenza. Despite promising in vitro and computational modelling results, these alternative inhibitors have not yet progressed to human clinical trials. Future research should prioritize investigating the therapeutic potential of dual-action inhibitors in clinical settings and explore other viral and bacterial proteins that contribute to co-infection dynamics, either by direct interaction or modification of the host environment. Baloxavir marboxil, an endonuclease inhibitor, has been found effective against oseltamivir-resistant influenza strains, offering a complementary approach in co-infection treatment [63]. Combination therapies, such as baloxavir with neuraminidase inhibitors, have also been explored and demonstrate enhanced viral suppression in preclinical studies, though clinical confirmation is still pending [64]. These emerging therapies, alongside dual-action neuraminidase inhibitors, represent promising strategies to reduce the severity of bacterial superinfections associated with influenza, ultimately improving patient outcomes.

Despite these findings, there are still notable gaps in our understanding of viral–bacterial co-infections. Most of the studies were conducted using animal models or in vitro systems, which may not fully capture the complexity of human co-infections. Moreover, very few Streptococcus species were investigated in these studies, and only four IAV subtypes were used. In addition, despite the well-documented synergistic effects of viral and bacterial proteins, further research is needed to clarify how these interactions affect host immune responses during co-infection.

This review is, to our knowledge, the first systematic review to focus specifically on the mechanistic interactions between viral and bacterial proteins during co-infections. Most previous reviews in this field have concentrated on clinical aspects of co-infections, such as case outcomes in influenza or COVID-19 patients [65,66], rather than the molecular and cellular mechanisms that underpin these interactions. This distinction highlights a key advantage of our review, as it provides valuable insights into the pathogenic pathways that could inform future therapeutic strategies.

## Figures and Tables

**Figure 1 pathogens-14-00114-f001:**
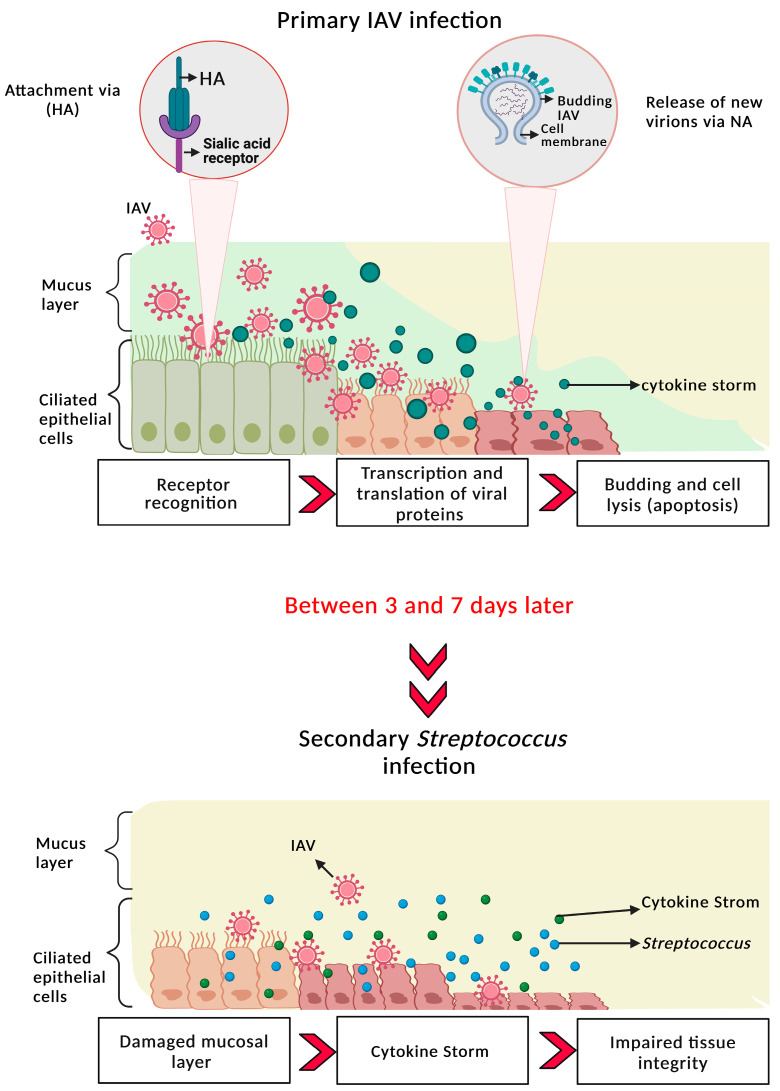
Illustration of primary IAV infection and secondary Streptococcus infection. The ciliated epithelial cells and underlying mucosal layers are damaged due to the sialidase activity of the viral NA. Viral infections impair the immune response, induce apoptosis, and cause inflammation, leading to tissue damage. This will enhance susceptibility to secondary bacterial infection 3–7 days post-viral infection. Created with BioRender.com.

**Figure 2 pathogens-14-00114-f002:**
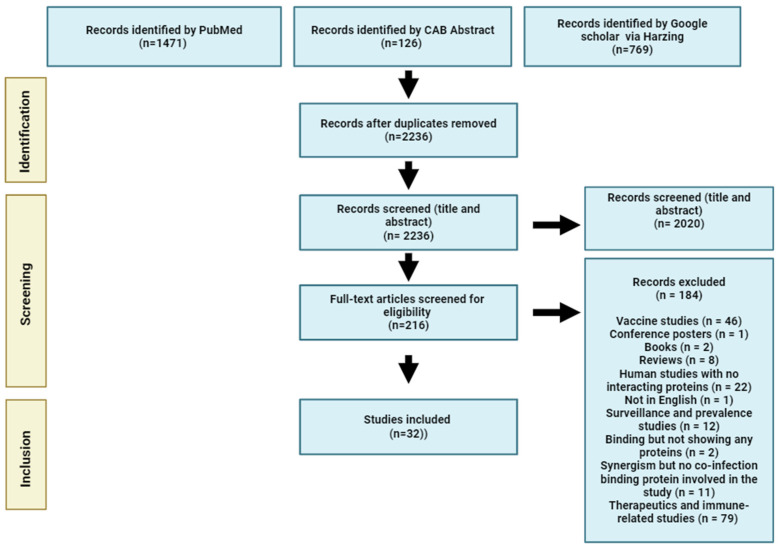
Flowchart summarising the process of the literature identification, screening, and selection for the scoping review.

**Figure 3 pathogens-14-00114-f003:**
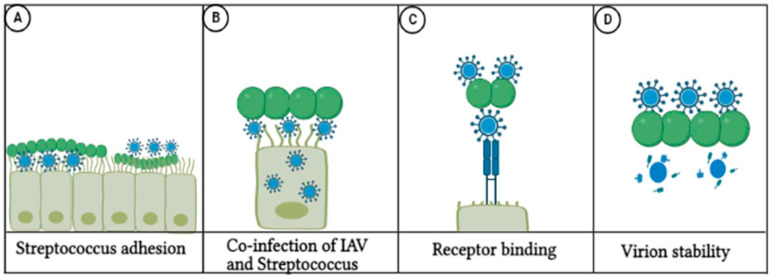
Direct interactions between Influenza A virus (IAV) and Streptococcus during co-infection. (**A**) Streptococcus adhesion to host epithelial cells, facilitated by viral factors during co-infection. (**B**) Co-infection of Influenza A virus (IAV) and Streptococcus, where viral particles enhance bacterial colonisation on host tissues. (**C**) Receptor binding between viral proteins and bacterial surface components, promoting bacterial adherence to host cells. (**D**) Virion stability is affected by bacterial interaction, which may facilitate the persistence of viral particles on host cells and enhance co-infection severity (Created with BioRender.com).

**Table 1 pathogens-14-00114-t001:** Overview of the characteristics of the 32 studies included in the analysis.

Study ID	Authors	Year	Influenza Virus	Bacteria	Model	Protein(s) Studied	Interaction	Key Finding
1	Ahmer, Raza et al. [40]	1999	Not stated	*Neisseria meningitidis Staphylococcus aureus* *Streptococcus pneumoniae Haemophilus influenzae*	HEp-2 cells	Viral: NA Bacterial: N/A	Indirect	NA activity from the influenza virus enhances bacterial binding to host cells. Bacterial adhesion increases significantly during influenza infection.
2	Okamoto, Kawabata et al. [36]	2004	A/Fort Monmouth/1/47 (H1N1)	*Streptococcus pyogenes strain* SSI-I (M3)	Mouse	Viral: HA Bacterial: N/A	Indirect	The HA of the influenza virus facilitates the internalisation of *S. pyogenes* into epithelial cells, leading to a lethal synergistic infection.
3	Platt, Lin et al. [33]	2022	A/Puerto Rico/8/1934 (H1N1)	*Streptococcus pneumoniae* (TIGR4)	A549 cells	Viral: HA, NA, NP Bacterial: Pneumolysin, Pneumococcal Surface Protein A	Indirect	Influenza A virus directly modulates *Streptococcus pneumoniae*, affecting bacterial growth, capsule production, and expression of virulence proteins.
4	Sakatani, Kono et al. [41]	2022	A HKx31 (H3N2)	Nonencapsulated *Streptococcus pneumoniae* (NESp MNZ11, MNZ1131 mutant)	Mouse	Viral: N/A. Bacterial: Pneumococcal Surface Protein K (PspK).	Indirect	PspK surface protein promotes colonisation and transmission of NESp. Influenza A virus co-infection enhances transmission, increasing both colonisation and shedding.
5	Sanford, Smith et al. [42]	1980	A/NWS/33 (H1N1) A/Puerto Rico/8/1934 (H1N1) A/Fort Monmouth/1/47 (H1N1) A/Japan/305/57 (H2N2) A/Texas/1/77 (H3N2)	Group B *Streptococcus* (types of Ia, Ib, Ic, II, III)	N/A	Viral: HA, NA Bacterial: N/A	Direct	HA of influenza A viruses facilitates adherence of virions to *Streptococcus*. Receptor-destroying enzyme (RDE) treatment destroyed virus adherence.
6	Ortigoza, Blaser et al. [26]	2018	A/X-31 (H3N2)	*Streptococcus pneumoniae*	Mouse	Viral: HA, NA. Bacterial: NanA and NanB	Indirect	NanA and NanB of *S. pneumoniae* antagonize IAV acquisition and transmission. Bacterial desialylation limits IAV infectivity.
7	Hafez, Abdel-Wahab et al. [34]	2010	H3N2	*Group Streptococcus pyogenes* (*M6*)	MDCK cells	Viral: HA Bacterial: M protein	Indirect	Viral infection increased adherence and internalisation of *S. pyogenes* into MDCK cells. Viral HA enhances interaction, while M protein is critical for adherence.
8	Herrera, Suso et al. [35]	2017	A/Hong Kong/1/68 (H3N2)	*Streptococcus pyogenes MGAS315* (*M3*)	Mouse	Viral: N/A Bacterial: M protein	Indirect	The M protein binds to host proteins (fibrinogen, albumin) during influenza infection, leading to enhanced bacterial virulence, lung barrier dysfunction, and mortality.
9	Wang, Gagnon et al. [43]	2013	A/swine/St-Hyacinthe/148/1990 (H1N1)	*Streptococcus suis* serotype 2 strain 31533	Tracheal epithelial cells	Viral: HA Bacterial: Capsular sialic acid	Direct	Capsular sialic acid of *S. suis* binds to swine influenza virus, enhancing bacterial adhesion and invasion of epithelial cells.
10	Chockalingam, Hickman et al. [44]	2012	H2N2 H9N2 A/Puerto Rico/8/1934 (H1N1)	*Streptococcus pneumoniae*	Mouse, A549 cells and mouse lung epithelial cells	Viral (s): Neuraminidase (NA) Stalk Region Bacterial: N/A	Indirect	NA stalk length of H2N2 and H9N2 does not affect pneumococcal adherence, but viral NA activity contributes to bacterial pneumonia, especially with oseltamivir treatment.
11	Nita-Lazar, Banerjee et al. [28]	2015	A/Puerto Rico/8/1934 (PR8)	*Streptococcus pneumoniae* (Sp3)	Mouse and A549 cells	Viral: HA, NA Bacterial: NA	Indirect	Galectin-1 and Galectin-3 bind to glycoproteins on PR8 virions. This enhances pneumococcal adhesion to desialylated airway epithelium. The viral NA activity increases galectin binding to host cells.
12	Grienke, Richter et al. [25]	2016	A/WSN/1933 (H1N1) A/Jena/8178/2009 (H1N1)	*Streptococcus pneumoniae*	MDCK and A549 cells	Viral: NA Bacterial: NA	Indirect	Prenylated flavonoids inhibited both viral and bacterial NAs, preventing synergism between influenza and *S. pneumoniae* and disrupting biofilm formation.
13	Walther, Xu et al. [27]	2016	A/WSN/1933 (H1N1), A/Jena/8178/2009 (H1N1)	*Streptococcus pneumoniae*	MDCK and A549 cells	Viral: HA Bacterial: NanA and NanB	Indirect	Both NanA and NanB enhance influenza virus replication by removing sialic acid receptors, allowing the virus to spread more easily. NA inhibitors (oseltamivir, artocarpin) were effective against both viral and bacterial NA.
14	Okamoto, Kawabata et al. [45]	2003	A/Fort Monmouth/1/47 (H1N1)	*Streptococcus pyogenes* (SSI-1, M1, M3)	Mouse	Viral: HA Bacterial: Streptococcal Pyrogenic Exotoxin A (SPEA)	Direct	Influenza A infection significantly enhances the internalisation of *S. pyogenes* into lung epithelial cells. Viral HA) expression promotes bacterial invasion, leading to a high mortality rate in mice.
15	Klenow, Elfageih et al. [23]	2023	A/Brisbane/02/2018 (H1N1) A/Kansas/14/2017 (H3N2)	*Streptococcus pneumoniae (TIGR4 strain)*	N/A	Viral: N1 and N2 Bacterial: NanA, NanB and NanC	Indirect	Both viral and bacterial NA enhance catalysis of sialic acid residues in the respiratory tract through distinct but similar mechanisms. The viral NA proteins are calcium-dependent, while bacterial ones are not.
16	Smith, Adler et al. [46]	2013	A/Puerto Rico/8/34 (H1N1) Recombinant PR8 with 1918 PB1-F2	*Streptococcus pneumoniae* (D39 and A66.1)	Mouse	Viral: PB1-F2 (1918 variant) Bacterial: NanA, NanB and NanC	Indirect	Co-infection significantly increases both viral titres and bacterial growth, leading to more severe lung pathology due to impaired macrophage function.
17	Wren, Blevins et al. [47]	2017	A/Puerto Rico/8/34 (H1N1)	*Streptococcus pneumoniae* EF3030, nanA mutant	Mouse	Viral: NA Bacterial: NanA	Indirect	*S. pneumoniae* NanA and viral NA synergistically enhance nasal colonisation and middle ear infection in the mouse model. Despite co-infection with IAV, NanA is crucial for maximal pneumococcal colonisation and infection, indicating that its role is not fully replaceable by the viral NA.
18	King, Lei et al. [48]	2009	A/Puerto Rico/8/34 (H1N1)	*Streptococcus pneumoniae* (D39, WU2, TIGR4)	Mouse	Viral: N/A Bacterial: Pneumococcal surface protein A (PspA), neuraminidase A (NanA), hyaluronidase (Hyl)	Indirect	PspA is essential for enhancing *S. pneumoniae* virulence during co-infection with influenza, and its absence significantly reduces bacterial survival. Immunization against PspA reduces bacterial burden and lung damage.
19	Wolf, Strauman et al. [49]	2014	A/Puerto Rico/8/34 (H1N1)	*Streptococcus pneumoniae* (*P1121 strain, serotype 23F, P1121ΔPLY mutant*)	Mouse	Viral: N/A Bacterial: Pneumolysin (PLY)	Indirect	Pneumolysin (PLY) expressed by *S. pneumoniae* protects colonised mice from influenza virus-induced morbidity and lung pathology. Prior colonisation with PLY-expressing *S. pneumoniae* modulates immune responses and reduces inflammation.
20	Egorova, Richter et al. [50]	2023	H3N2, H1N1	*Streptococcus pneumoniae* (NanA-expressing strains)	MDCK cells	Viral: NA Bacterial: NanA	Indirect	Pyrrolo[2,3-e]indazole compounds inhibit both influenza A and pneumococcal NA, showing dual activity and potential for co-infection treatment.
21	McCullers and Rehg [51]	2003	A/Puerto Rico/8/34 (H1N1)	*Streptococcus pneumoniae* (D39 strain)	Mouse	Viral: NA Bacterial: N/A	Indirect	Viral NA enhances pneumococcal adherence by stripping sialic acid, promoting bacterial colonisation. Oseltamivir reduced mortality by inhibiting viral NA, limiting pneumonia.
22	Dutta, Chen et al. [9]	2021	A/Puerto Rico/8/34 (H1N1)	*Streptococcus pneumoniae* (serotype 3, Taian strain, wild type and NanC-inserted mutant)	Mouse	Viral: N/A Bacterial: NanA, NanB and NanC	Indirect	NanC reverses NanA/NanB-mediated suppression of anti-influenza immunity, improving virus clearance but worsening inflammation and lung pathology.
23	Wu, Meng et al. [52]	2015	Swine H1N1 Swine H3N2 A/Puerto Rico/8/34 (H1N1)	*Streptococcus suis* serotype 2 (wild type and non-capsulated mutant)	Porcine tracheal cells	Viral: HA Bacterial: N/A	Direct	SIV and *Streptococcus suis* directly interact through viral HA binding to sialic acid on the bacterial capsule. This binding enhances bacterial adherence and invasion of porcine tracheal cells, but co-infection decreases viral replication.
24	Park, Gonzalez-Juarbe et al. [53]	2021	A/Puerto Rico/8/34 (H1N1)	*Streptococcus pneumoniae* (EF3030 strain and ΔpspA mutant)	Mouse	Viral: N/A. Bacterial: Pneumococcal surface protein A (PspA), Pneumolysin	Indirect	*S. pneumoniae* uses PspA to bind to GAPDH on dying epithelial cells, enhancing bacterial adherence and worsening pneumonia after influenza infection. PspA-GAPDH binding is enhanced by pneumolysin and IAV infection.
25	Herrera, Faal et al. [22]	2018	A/Hong Kong/1/68 (H3N2)	*Streptococcus pyogenes* (GAS, MGAS315 strain, PrtF.2 mutant)	Mouse and A549 cells	Viral: N/A. Bacterial: Pneumococcal surface protein A (PspA), Pneumolysin	Indirect	PrtF.2 mutant of *S. pyogenes* contributes to bacterial adherence and virulence by binding to tenascin C (TNC) on host cells. Viral NA indirectly facilitates this binding by exposing TNC and fibronectin after removing sialic acid. Deleting PrtF.2 reduces bacterial adherence and virulence.
26	Herrera, Van Hove et al. [38]	2020	A/Puerto Rico/8/34 (H1N1)	*Streptococcus pyogenes* (MGAS315, M3 serotype)	Mouse	Viral: N/A. Bacterial: M protein and Streptolysin O (SLO)	Indirect	Passive immunotherapy using antisera targeting the M protein or streptolysin O (SLO) reduced morbidity in IAV- *S. pyogenes* superinfection but did not significantly reduce mortality. Antisera improved *S. pyogenes* clearance from lungs but had a limited effect on survival.
27	Weeks-Gorospe, Hurtig et al. [54]	2012	A/Puerto Rico/8/34 (H1N1)	*Streptococcus pneumoniae Staphylococcus aureus Streptococcus pyogenes*	Mouse	Viral: PB1-F2. Bacterial: N/A	Indirect	Full-length PB1-F2 variants with specific amino acids (62L, 66S, 75R, 79R, 82L) were linked to increased bacterial virulence. Truncated PB1-F2 exacerbated mortality in *S. pyogenes* infections. PB1-F2 modulates the host immune response, increasing bacterial load and mortality during co-infection.
28	Klonoski, Watson et al. [55]	2018	A/Puerto Rico/8/34 (H1N1) A/swine/Texas/4199-2/98 (H3N2) reassortant strains	*Streptococcus pyogenes* (MGAS315)	Mouse	Viral: HA, PB1, PB2, NS1 Bacterial: N/A	Indirect	The viral HA, PB1, and NS gene segments of the influenza virus contribute to the severity of bacterial superinfection by modulating immune responses. PR8-derived PB1 and NS1 genes worsened outcomes, increasing mortality in infected mice.
29	Li, Ren et al. [24]	2015	A/Puerto Rico/8/34 (H1N1)	*Streptococcus pyogenes Streptococcus pneumoniae Staphylococcus aureus Haemophilus influenzae*	Mouse and invitro A549 cell line	Viral: NA Bacterial: Fibronectin-binding proteins	Indirect	Influenza NA activates TGF-β, upregulating host cell receptors such as fibronectin (Fn) and α5 integrin, which facilitate bacterial adherence. Inhibiting TGF-β signalling reduces bacterial attachment.
30	Okamoto, Kawabata et al. [56]	2004	A/Fort Monmouth/1/47 (H1N1)	*Streptococcus pyogenes* (wild type M1 strain SSI-9, and capsule mutants)	Mouse and A549 cells	Viral: HA Bacterial: Capsular hyaluronic acid, M protein, Streptolysin O (SLO) Streptolysin S (SLS)	Direct	The capsule of *S. pyogenes* is essential for adherence to virus-infected epithelial cells and induces lethality in bacterial-viral superinfection. Capsule mutants showed significantly reduced bacterial adhesion and mortality. Wild-type bacteria directly bind to influenza virus particles on infected cells.
31	Hoffmann, Richter et al. [57]	2017	A/Jena/5258/2009 (HXNX) A/Jena/8178/2009 (HXNX) A/WSN/1933 (H1N1) A/Hong Kong/68 (H3N2)	*Streptococcus pneumoniae* (serotype 1 DSM20566)	N/A	Viral: NA Bacterial: NanA	Indirect	Diazenylaryl sulfonic acids were identified as dual inhibitors of viral and bacterial NA, effective at low micromolar concentrations. The most active compound, NSC65847, exhibited mixed-type inhibition against both viral and bacterial NA, showing potential as a dual-acting anti-infective.
32	Peltola, Murti et al. [58]	2005	A/Puerto Rico/8/34 (H1N1) A/Hong Kong/1/68 (H3N2) A/Singapore/1/57 (H2N2) A/England/12/62 (H2N2) A/Memphis/102/72 (H3N2) A/Leningrad/516/86 (H3N2) A/Sydney/5/97 (H3N2) A/Chicken/Hong Kong/WF2/99 (H9N2)	*Streptococcus pneumoniae* (lux-transformed D39, strain R6T)	Mouse and A549 cells	Viral: NA Bacterial: N/A	Indirect	NA activity enhances bacterial adherence to respiratory epithelial cells, promoting secondary bacterial pneumonia. The higher NA activity correlates with increased bacterial adherence and higher mortality in co-infected mice. Oseltamivir reduced bacterial adherence by inhibiting NA.

## Data Availability

Data will be made available by the corresponding author if requested.

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
