# Peer review of "Understanding the Molecular Interactions Between Influenza A Virus and Streptococcus Proteins in Co-Infection: A Scoping Review"

_pathogens, 2025, doi:10.3390/pathogens14020114_

Round 1

Reviewer 1 Report

Comments and Suggestions for Authors

Reviewer Comments

1. Can the authors explain the significance of examining the IAV replication and Secondary bacterial infections in pigs to provide context for the reader?

2. Line 371: This observation aligns with findings reported in numerous prior publications. It would be beneficial to include a few references here or in the discussion to support this correlation.

3. The primary focus was understandably on IAV. However, did the authors also investigate the presence of other S. pneumoniae-related protein interactions and related therapeutic? Clarifying or discussing this would be valuable.

4. Line 216: Please clarify the definition of co-infection. Does it refer to two different pathogens within the same category (e.g., viruses) or pathogens from different categories?

Author Response

Reviewer comment 1. Can the authors explain the significance of examining the IAV replication and secondary bacterial infections in pigs to provide context for the reader?

Author response: We have added the following paragraph to the introduction (lines 39–47):

IAV infection can also cause respiratory disease in both pigs and horses. Uncomplicated IAV infection is usually fairly mild in pigs but co-infection with other pathogens can cause severe disease. Pigs are highly susceptible to secondary bacterial infections during infection with IAV [1-3] and thus a good model for studying the interaction between viral and bacterial pathogens. Pigs may also serve as a vessel for IAV evolution and emergence of new stains which can be zoonotic [1]. The pathogenesis of IAV-induced immune suppression and epithelial damage in pigs can provide important insights into the pathogenesis of these co-infections, which are pertinent to both human and veterinary health [1].

Reviewer comment 2.  Line 371: This observation aligns with findings reported in numerous prior publications. It would be beneficial to include a few references here or in the discussion to support this correlation.

This refers to the sentence: "Viral NA and bacterial NA (NanA) worked together to increase nasal colonisation and exacerbate the severity of middle ear infections in the mouse model."

Author response: There are indeed many papers discussing how viral and bacterial NAs increase bacterial colonisation and exacerbate middle ear infections (in humans). However, we excluded these references because this section specifically discusses results from the systematic review which excluded human studies. However, to address the reviewer's concern, we have added the following paragraph in the discussion (lines 472–478):

Despite the fact that this review focuses on non-human cases and excludes human studies, similar mechanisms have also been documented in human infections. Specifically, studies on human co-infections have shown that viral and bacterial NAs cooperate to enhance bacterial colonisation and exacerbate disease severity, including middle ear infections [65]. However, our focus on animal studies aimed to provide a targeted understanding of these interactions in non-human models, as they offer unique insights into the dynamics of viral and bacterial protein interactions under experimental conditions.

Reviewer comment 3. The primary focus was understandably on IAV. However, did the authors also investigate the presence of other S. pneumoniae-related protein interactions and related therapeutic? Clarifying or discussing this would be valuable.

Author response: Regarding this point, we are not entirely sure we understand the reviewer’s concern. I’ve already addressed the investigation of other S. pneumoniae-related protein interactions and their therapeutic relevance in the Bacterial Proteins section (3.2).

To make this clearer, we have added the following explanation in the discussion (lines 466–473):

In addition to viral NA and HA, this review highlights not only the extensively studied bacterial neuraminidase proteins (NanA, NanB, and NanC) but also other key S. pneumoniae-related proteins such as PLY, PspA, and PspK. These proteins are critical in facilitating bacterial adhesion, colonisation, and immune evasion during viral-bacterial co-infections. Additionally, the role of proteins like M protein (S. pyogenes), streptolysin O, and streptolysin S demonstrates that similar virulence mechanisms exist across different Streptococcus species, highlighting potential therapeutic targets across bacterial pathogens.

Reviewer comment 4. Line 216: Please clarify the definition of co-infection. Does it refer to two different pathogens within the same category (e.g., viruses) or pathogens from different categories?

Author response: To address this comment, we have added the following clarification in the manuscript (lines 227–231):

Co-infections, defined as concurrent infections caused by two or more pathogens, can involve either pathogens from the same category (e.g., multiple viruses) or from different categories (e.g., viruses and bacteria), and are a significant cause of increased morbidity and mortality compared with uncomplicated infections. In this review, the term co-infection specifically refers to cases where viral and bacterial pathogens interact to cause disease.

Reviewer 2 Report

Comments and Suggestions for Authors

This manuscript addresses an important area of infectious disease research by focusing on molecular interactions during viral-bacterial coinfections. It provides valuable insights into the molecular mechanisms underlying viral-bacterial coinfections, which are crucial for understanding the outcomes of severe human diseases.

In the “Methods” section, the manuscript thoroughly explains the criteria for data selection. However, it would be more convincing if it also discussed the potential impacts of not adhering to these criteria. In the “Discussion” section, the manuscript could compare this study with similar research, highlighting its advantages and limitations.

Overall, the manuscript effectively summarizes the current understanding of the molecular interactions and infection processes during the coinfection of influenza virus and Streptococcus.

Author Response

Comment 1: This manuscript addresses an important area of infectious disease research by focusing on molecular interactions during viral-bacterial coinfections. It provides valuable insights into the molecular mechanisms underlying viral-bacterial coinfections, which are crucial for understanding the outcomes of severe human diseases.

Response 1: The authors thank the reviewer for their positive comment.

Comment 2: In the “Methods” section, the manuscript thoroughly explains the criteria for data selection. However, it would be more convincing if it also discussed the potential impacts of not adhering to these criteria.

Response 2: We have added the following sentences to methods (lines 146-151):

To ensure the inclusion of high-quality and relevant studies, strict adherence to these criteria was essential to preserve the reliability of the findings. Any deviation from these criteria could have introduced biases, undermined the study’s validity, or resulted in the inclusion of studies with limited methodological rigour. For example, studies with poorly defined endpoints or inappropriate populations could have weakened the strength of the conclusions drawn.

Comment 3: In the “Discussion” section, the manuscript could compare this study with similar research, highlighting its advantages and limitations.

Response 3: The authors are not aware of any prior systematic reviews looking at specifically at mechanisms of viral-bacterial co-infections and pathogenesis. Published systematic reviews in this area have studied clinical cases (influenza, COVID etc), rather than mechanisms.

Added the following sentences to the discussion (lines 527-533) to emphasise this:

This review is, to our knowledge, the first systematic review to focus specifically on the mechanistic interactions between viral and bacterial proteins during co-infections. Most previous reviews in this field have concentrated on clinical aspects of co-infections, such as case outcomes in influenza or COVID-19 patients [64, 65], rather than the molecular and cellular mechanisms that underpin these interactions. This distinction highlights a key advantage of our review, as it provides valuable insights into the pathogenic pathways that could inform future therapeutic strategies.

Comment 4: Overall, the manuscript effectively summarizes the current understanding of the molecular interactions and infection processes during the coinfection of influenza virus and Streptococcus.

Response 4: The authors thank the reviewer for their positive comment.

Reviewer 3 Report

Comments and Suggestions for Authors

The authors searched all the works that study the mechanism behind the superinfection of Influenza A virus and Streptococcus bacteria. I believe this review may benefit the ones that are interested in this field. I only have a few minor questions and suggestions:

Line 88-91 mentions how LPS could be used for immune evasion, but Streptococcus is gram-positive and LPS is usually a hallmark of gram-negative bacteria. Does Streptococcus have LPS?

The functions of the bacterial proteins discussed in section 3.2 seem to be independent of viral infection, thus irrelevant to this review.

Line 252-253 states that viral HA may directly bind to sialic acid of bacterial surface, which is not shown in Figure 1. Also, it is suggested later that viral HA promotes ‘the removal of sialic acid’, which I assume is the sialic acid on the host membrane, meaning viral HA could recognize sialic acid from both Streptococcus and the host unbiasedly. Does viral HA directly bind to bacterial sialic acid with no cleavage, but remove host sialic acid? How does IAV achieve this specificity? Is there any reference addressing this question?

It is hard to understand line 269-271. What does it mean by Streptococcus suis ‘binding to bacterial sialic acid’?

What the authors discuss in the majority of section 3.4 is essentially within the same mechanism. Therefore, it is not quite correct to split the content into several paragraphs, indicating that different mechanisms exist in the interplay between viral and bacterial neuraminidases.

Author Response

Comment 1: Line 88-91 mentions how LPS could be used for immune evasion, but Streptococcus is gram-positive and LPS is usually a hallmark of gram-negative bacteria. Does Streptococcus have LPS?

Response 1: Apologies for this error, the reviewer is correct in that there is no LPS in Streptococcal bacteria - the confusion arose due to citing a general review. The sentence has been edited to: "In complex co-infections, the released sialic acids are also broken down for metabolic use as a carbon source by bacteria, while modifications to bacterial surface molecules (which for gram negative bacteria include lipopolysaccharides (LPS)), help bacteria evade the host's immune system, form protective biofilms, and resist environmental stresses" (lines 88-92).

Comment 2: The functions of the bacterial proteins discussed in section 3.2 seem to be independent of viral infection, thus irrelevant to this review.

Response 2: The bacterial mechanisms highlighted in this section have been shown in a number of cited studies to be important in the pathogenesis of influenza-Streptococcal co-infections. For example, Streptococcal PspA facilitates bacteria binding to host cells that have been previously damaged by influenza infection which leads to a worsening of the secondary bacterial infection (Ref 52). The reviewers make a valid comment that these proteins may function in the absence of viral infection, however, the cited papers highlight their importance in co-infections and the authors feel that the inclusion of this information is important (for example in determining treatment for viral-bacterial co-infections).

Comment 3: Line 252-253 states that viral HA may directly bind to sialic acid of bacterial surface, which is not shown in Figure 1.

Response 3: We thank the reviewer for highlighting our error. This has been corrected to refer to Figure 3 (line 264).

Comment 4: Also, it is suggested later that viral HA promotes ‘the removal of sialic acid’, which I assume is the sialic acid on the host membrane, meaning viral HA could recognize sialic acid from both Streptococcus and the host unbiasedly. Does viral HA directly bind to bacterial sialic acid with no cleavage, but remove host sialic acid? How does IAV achieve this specificity? Is there any reference addressing this question?

Response 4: Sorry, this was a typographical error and should have referred to viral NA which has now been amended (line 261). This section focused on the role of viral HA and non-structural proteins, and included a total of 12 studies. Of these, 5 studies demonstrated direct interactions involving HA binding to bacterial components like sialic acid. The remaining 7 studies focused on indirect interactions, where the host environment is altered by the action of viral NA to facilitate bacterial colonisation.

Comment 5: It is hard to understand line 269-271. What does it mean by Streptococcus suis ‘binding to bacterial sialic acid’?

Response 5: This was line 279 in the downloaded manuscript: "also revealed a direct interaction between viral HA and Streptococcus suis, with the latter binding to bacterial sialic acid to promote bacterial adherence and invasion of pig tracheal cells while suppressing viral replication." This sentence should have been deleted (as it was replaced by the rephrased previous sentence) - and has now been deleted, thank you.

Comment 6: What the authors discuss in the majority of section 3.4 is essentially within the same mechanism. Therefore, it is not quite correct to split the content into several paragraphs, indicating that different mechanisms exist in the interplay between viral and bacterial neuraminidases.

Response 6: Paragraphs are used merely as an aid to improve structure and "read-ability", not to refer specifically to different mechanisms.

We thank the reviewer for a detailed assessment of our manuscript and for allowing us to correct the highlighted errors.